# Effect of storage temperature on the dispersibility of commercially available 0.1% fluorometholone ophthalmic suspension

**Tokio Obata**[1], **Saori Deguchi**[2], **Jyoji Yoshitomi**[2], **Kazunori Inaba**[2], **Yoko Urashima**[1], **Takuro Kobori**[1], **Kouichi Hosomi**[2], **Noriaki Nagai**[2], **Yuichiro Nakada**[1]*

1 Faculty of Pharmacy, Osaka Ohtani University, Tondabayashi, Osaka, Japan, 2 Faculty of Pharmacy, Kindai University, Higashi-Osaka, Osaka, Japan

* nakadayu@osaka-ohtani.ac.jp

## Abstract

In this study, we focused on the storage conditions and investigated the effects of low-temperature storage (10˚C) on the dispersibility of active components in three formulations of fluorometholone (FLU) suspension eye-drops (one original drug and two generic drugs, P1-P3). For all three eye-drop products, before shaking by hand, white sediment anticipated to be the principal active component was seen at the vial base. In the ordinary-temperature storage group, the FLU contents per drop after shaking by hand were 0.076% in P1, 0.023% in P2, and 0.100% in P3, and the content in P2 was significantly lower than that in P1 and P3. In contrast, almost no dispersion was observed in the low-temperature group. The results after sufficient shaking of these samples with a vortex, in contrast, were such that the FLU contents per drop were 0.063% in P1, 0.086% in P2, and 0.088% in P3; the content in P1 was significantly lower than that in P2 and P3, and there was no difference between P2 and P3. Moreover, we evaluated the dispersibility according to the evaluation "$V_s / (\rho_g − \rho_f)$ g." In both the low- and ordinary-temperature storage groups, the value of $V_s / (\rho_g − \rho_f)$ g, proportional to the terminal velocity, decreased in the following order: P3 > P1 ≫ P2, and each value in the ordinary-temperature was higher than that in low temperature. The zeta potential decreased in the following order: P2 > P3 ≫ P1. In conclusion, when FLU suspension eye drops are stored at low temperatures until use, such as in a refrigerator, ordinary shaking does not help achieve dispersion to the specified concentration, and even with vigorous shaking with some formulations, the specified concentration cannot be achieved.

## Introduction

Many drugs incorporated in eye-drops have low water solubility, and in such cases, aqueous-suspension eye-drops and eye ointments are considered the ideal dosage forms [1]. These aqueous-suspension eye-drops are aqueous formulations prepared by suspending poorly soluble drugs as fine crystals or fine powder in aqueous media. Eye-drops have a simpler application method than eye ointments and involve minor discomfort when applied. In contrast, suspension formulations are associated with problems such as the growth of dispersed

**Data Availability Statement:** All relevant data are within the manuscript.

**Funding:** The authors received no specific funding for this work.

**Competing interests:** The authors have declared that no competing interests exist.

particles, and thus, an increase in particle diameter during storage affects the biological activity [2]. It is also reported that the aggregates adhere to the containers. Destabilization of such formulations is considered to cause changes in the drug content per drop of the eye-drop product, which in turn affects the efficacy. It is therefore essential to instruct patients to shake eye-drops sufficiently before use.

Fluorometholone (FLU) suspension eye-drops (FLU eye-drop) are one of the suspension eye-drops mentioned above. The storage temperature for FLU eye-drops is set to room temperature (1–30 ˚C) in Japan and 2–25 ˚C in the United States [3–10]. As a caution relating to the storage of FLU eye-drops, Nitto Medic Co. (Toyama, Japan) released a statement in 2012 suggesting that storage in cold places such as refrigerators should be avoided [11], and the storage temperature is known to affect the physical properties of FLU eye-drops. We previously investigated the effect of the storage duration of original and generic drugs for FLU eye-drops on the drug content per drop and their physical stability (dispersibility and particle diameter) after mixing by shaking by hand. With several generic products, it was shown that, unlike the original drugs, gentle mixing does not allow the drug concentration to reach the recommended levels [1]. These findings suggest that storage of FLU eye-drops has a significant effect on their dispersion stability and thus on drug delivery during administration. Considering recent Patient-Focused Drug Development in the USA [12], it is suggested that if it is challenging to develop formulations that aim to eliminate the predictable disadvantages in patients, it is vital to provide information about the problems associated with the products. In addition, eye-drops are indispensable for treatment in ophthalmology, and their use by patients has major effects on their therapeutic efficacy. Furthermore, decreased adherence by the patients sometimes results in discontinuation of eye-drop treatment; hence, the provision of providing instructions by pharmacists or other healthcare professionals is crucial. Nevertheless, according to a report by Ono et al., in a questionnaire-based survey of 200 patients admitted for the first time to the Dept. of Ophthalmology at Oita University's Faculty of Medicine, performed between May 1, 2018, to February 28, 2019, more than half, i.e., 56%, of subjects responded that they had received no instructions regarding the use of eye-drops. This suggests that instructions about the proper use of eye-drops are not currently being provided [13]. It is, therefore, possible that appropriate instructions about the storage methods are also not being offered.

The current study focused on FLU eye-drops, and, in order to clarify the caution regarding the storage method needed to be included in instructions for use, the effects of low-temperature storage on the dispersibility of the active component were investigated in detail with the original drug and two generic drugs.

## Materials and methods

### Materials

A total of three commercially available FLU eye-drop products were used, these being the original drug, Product-1 (Flumetholon ophthalmic suspension 0.1%, Santen Pharmaceutical Co., Ltd., Osaka, Japan: P1), and two generic products, Product-2 (Fluorometholone Ophthalmic Suspension 0.1% [NITTO], Nitto Medic Co., Ltd., Toyama, Japan: P2) and Product-3 (FLUOROMETHOLONE ophthalmic solution 0.1% [WAKAMOTO], WAKAMOTO PHARMACEUTICAL CO., LTD., Tokyo, Japan: P3) (Table 1). Each product was divided into two groups. One group was named the low-temperature storage group, where the product was stored for 3 months at low temperature (10˚C), and subsequently, for 2 months at normal room temperature, while the other group was the ordinary-temperature storage group, stored

**Table 1. Product characteristics of 0.1% FLU ophthalmic suspensions used in this study.**

| Formulation | Additives | Material of the bottle |
|---|---|---|
| P1 | Sodium edetate hydrate, Sodium chloride, Benzalkonium chloride, Sodium dihydrogen phosphate hydrate, Polysorbate 80, Methylcellulose, Sodium hydrogen phosphate hydrate | Polyethylene |
| P2 | Sodium chloride, Benzalkonium chloride, Polysorbate 80, Sodium edetate hydrate, Sodium hydrogen phosphate hydrate, Sodium dihydrogen phosphate hydrate, Polyvinyl alcohol (partially saponified) | Polyethylene |
| P3 | Potassium dihydrogen phosphate, Sodium hydrogen phosphate hydrate, Sodium chloride, Benzalkonium chloride, Methylcellulose | Polypropylene |

for 5 months at normal room temperature. The storage materials remained undisturbed and in the upright position.

## Shaking of eye-drops

Each of the eye-drop products was shaken by hand and by using a vortex mixer. For shaking by hand, the vial was grasped with the thumb and index fingers, and, with the elbow remaining motionless, the lower arm was moved 10 times from 45 to 90˚, at one move per second (Fig 1). Shaking with the vortex involved sufficient shaking for 1 min. From each of the shaken eye-drop vials, one drop was collected for use as the measurement sample. In addition, in order to avoid the effect of bubbles when stirring with a vortex, separated from the above shaking, a vial was shaken gently enough such that no bubbles were formed on the day after the vortex shaking. Furthermore, a sample was collected, and the FLU concentration was measured.

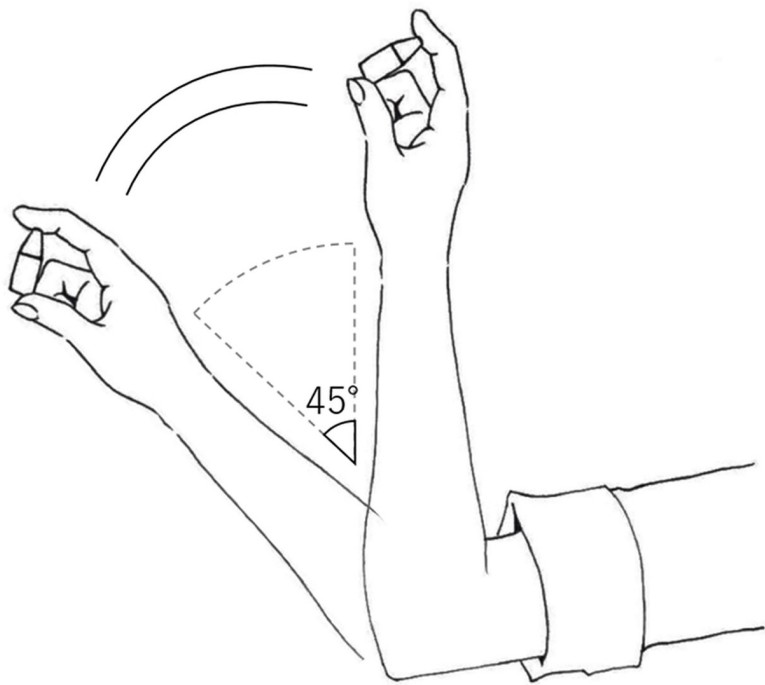

**Fig 1. The method used for agitation with hands.** Each bottle of 0.1% FLU ophthalmic suspension was held between the thumb and index finger at a standing position, and subsequently, the elbow was placed on the table. Subsequently, each bottle was shaken at a given angle ranging from 45–90 degrees every second, approximately 10 times.

## Confirmation of dispersibility

To confirm dispersibility, the eye-drops were shaken by hand as detailed under the sub-heading 2 ("Shaking of eye-drops"), and the characteristics of the particles in one drop dripped out of the vial were observed using a light microscope (BA210E; Shimadzu Corporation, Kyoto, Japan), and evaluated. In addition, the drug particle distribution and mean particle diameter were measured using laser diffraction equipment (SALD-7100; Shimadzu Corporation), with a refractive index of 1.60 to 0.10. Furthermore, the FLU content per drop of the eye-drop was measured by high-performance liquid chromatography (HPLC). Each test was performed at least three times, and reproducibility was confirmed.

## FLU concentration measurement

The FLU content in one drop of each of the eye-drop products was measured by HPLC. To 20 μL of the sample, 50 μL of the internal standard, 50 μg/mL p-hydroxybenzoate n-butyl in methanolic solution was added, and the resulting mixture was filtered using a Chromatodisk 4A (pore size: 0.45 μm; Kurabo Industries, Ltd., Osaka, Japan), and injected into the HPLC device (LC-20AT; Shimadzu Corporation). A TSK gel ODS-100V column (5 mm; internal diameter: 4.6 mm; length: 15 cm; Tosoh Bioscience, Inc., Tokyo) was used after equilibration with a mobile phase that was a 7:3 methanol/water mixture. The volume injected into the HPLC device was 10 μL, and the measurement temperature was maintained at 35˚C using an oven column (CTO-20A; Shimadzu Corporation). The flow speed of the mobile phase was 0.8 mL/min, and ultraviolet absorption was measured at 254 nm. In this study, the FLU detection time was 12.5 min, and the following favorable calibration curve was $y = 0.0012x + 0.0272$ ($R^2 = 0.9963$). The lower limit of the quantification of the FLU concentration was 50 μg/mL.

## Evaluation of physical properties of each eye-drop product

The viscosity of each eye-drop product was measured using a tuning-fork vibration viscometer (SV-1A; A&D Co., Ltd., Tokyo, Japan) at temperatures ranging from 6 to 40˚C. The zeta potential was measured using a Micro-Electrophoresis Zeta Potential Analyzer Model 502 (Sanyo Trading Co., Ltd., Tokyo, Japan). The DTA of each eye-drop product and FLU-based powder was performed using a simultaneous thermogravimetry-differential thermal analysis device (DTG-60H; Shimadzu Corporation).

## Analysis of the FLU sedimentation process in eye-drops using Stokes' equation

Since the particles that undergo sedimentation are no more than 100 μm in diameter, the terminal velocities of particles that were sedimented in each eye-drop product were compared using Stokes' equation (Eq 1), as follows:

$$V_s = D_p^2 \left( \rho_g - \rho_f \right) g \, / \, 18\eta \tag{1}$$

Where $V_s$ is terminal velocity, $D_p$ is particle diameter (m), $\rho_g$ is particle density (kg/m$^3$), $\rho_f$ is fluid density (kg/m$^3$). g and $\eta$ are gravitational acceleration (m/s$^2$) and fluid viscosity (Pa·s), respectively.

Moreover, Eq 2, a modified version of Eq 1 was also used in the analysis:

$$V_s / (\rho_g - \rho_f) \, g = D_p^2 / 18\eta \tag{2}$$

In this case, it is assumed that there are no differences in FLU particle density between the eye-drop products and that the fluid density of suspension eye-drops is approximately similar to that of water, because of the low FLU concentration (0.1%); the left side of Eq 2 is simply proportional to the particle terminal velocity. In other words, as the terminal velocity is affected by particle diameter and fluid viscosity, it was analyzed that it could be due to the sedimentation condition in the current study.

## Measurement of FLU adhesion to eye-drop vials

Considering P1, P2, and P3 as samples for the low-temperature storage group, they were shaken vigorously using the vortex, after which the entire volume was taken, and each of the vials was washed with 1 mL of purified water. After washing, 1 mL of methanol was added to the eye-drop vial. The sample was prepared by shaking for 1 h with 360˚ rotation, and the concentrations were measured by HPLC. This shaking was performed using a revolution mixer (RVM-101; Synix Inc., Tokyo, Japan), and a total of 10 rotations were performed at the speed of 1 rotation per 3 s.

## Statistical analysis

Data are expressed as means ± standard deviations. The software used for statistical analysis was GraphPad Prism 3 (GraphPad Software, Inc.; La Jolla, California, USA). The tests performed were one-way analysis of variance, and Tukey's multiple comparison test, with differences considered statistically significant at $p < 0.05$.

## Results

### Shaking of eye-drop vials did not abolish the deposition of flu caused by low temperature-storage

Fig 2 shows the basal part of a vial in the low-temperature storage group after shaking by hand. For all three of the eye-drop products included in this study, before shaking by hand (Pre), white sediment, likely the principal active component, was observed at the base of the vial. After shaking by hand (Post), this sediment remained at the base of the vial, showing almost no decrease in the sediment bed. Fig 3 shows the DTA of P1, P2, and P3 in the low- and ordinary-temperature storage groups. The curves were approximately the same for each of the eye-drop products in each of the storage groups, and there were no marked differences in the melting point peak and no differences as compared to the FLU-based powder.

### Dispersibility of FLU ophthalmic suspensions is dependent on the storage temperature and shaking methods

Fig 4 shows a typical FLU particle in one drop of each sample after shaking by hand, and Fig 5 shows the drug particle distribution in each sample measured using a laser diffraction particle size analyzer. The particles appeared to disperse with all samples in the ordinary-temperature storage groups, whereas almost no dispersion was observed in the low-temperature group (Fig 4). Similarly, in the investigation using the laser diffraction particle distribution measurement device, dispersion of particles was observed at ordinary-temperature storage (Fig 5), whereas no particles were detected in the low-temperature storage group. Fig 6 shows the FLU contents per drop after shaking by hand. In the ordinary-temperature storage group, the FLU contents per drop after shaking by hand were 0.076 ± 0.010% in P1, 0.023 ± 0.006% in P2, and 0.100 ± 0.005% in P3; the content in P2 was significantly less than those in P1 and P3. In the low-temperature storage group, in contrast, the content in P1 was 0.001 ± 0.001%, and the

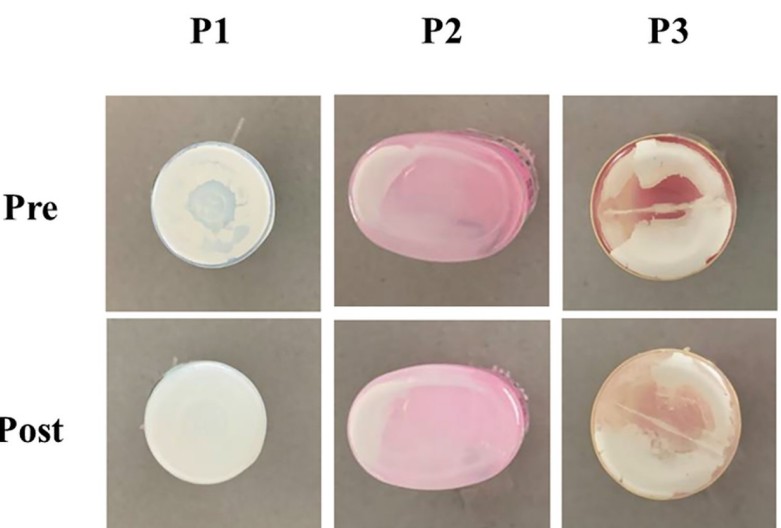

**Fig 2. Changes in the deposition of 0.1% FLU ophthalmic suspensions attached to the underside of each container before and after agitation with hands.** All the formulations of 0.1% FLU ophthalmic suspensions were stored at 10 ˚C for three months and subsequently maintained at room temperature for two months. Optical images for the deposition of FLU ophthalmic suspensions attached to the underside of each container before and after agitation with hands are shown. P1; Original, P2; Generic A, P3; Generic B. All the images are representative of three independent experiments.

contents in P2 and P3 were below the limits of quantification. When shaken by hand, the FLU content per drop was markedly lower in the low-temperature storage group relative to the ordinary-temperature storage group. The results after sufficient shaking of these samples with a vortex, in contrast, were such that the FLU contents per drop were 0.063 ± 0.011% in P1, 0.086 ± 0.001% in P2, and 0.088 ± 0.001% in P3; the content in P1 was significantly lower than

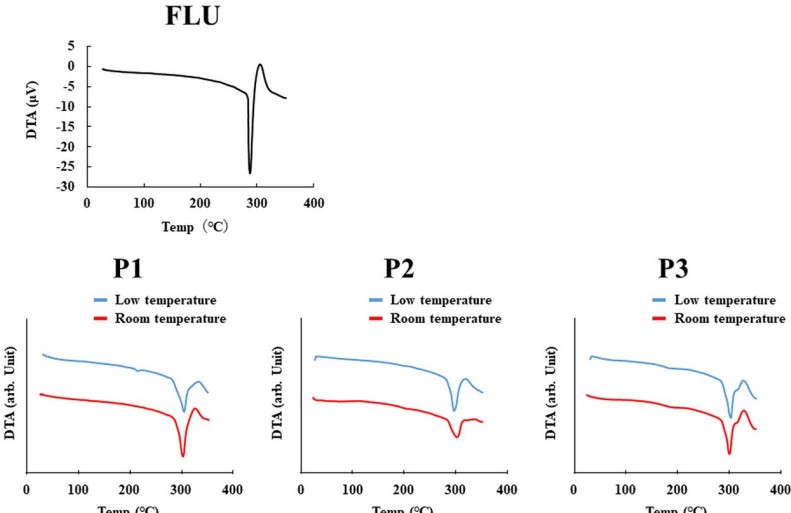

**Fig 3. Changes in the melting points of each formulation of 0.1% FLU ophthalmic suspensions.** In low (blue line) and room (red line) temperature groups, each formulation of 0.1% FLU ophthalmic suspension was stored at 10 ˚C for three months and subsequently maintained at room temperature for two months or left at room temperature for five months. The melting point of each formulation was measured on a simultaneous thermogravimetric analyzer. P1; Original, P2; Generic A, P3; Generic B, DTA; Differential thermal analysis.

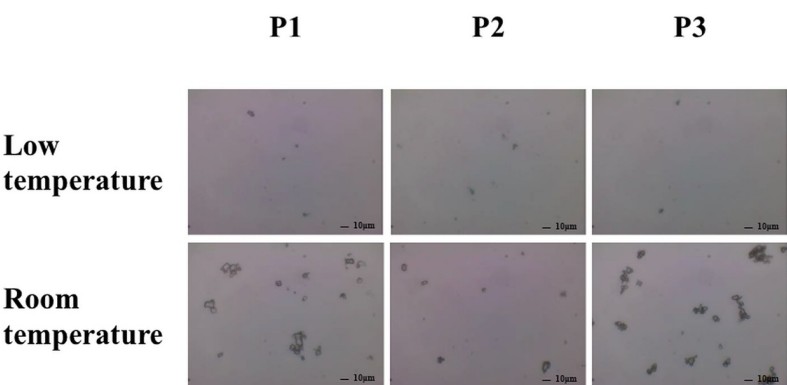

**Fig 4. Influence of storage temperature on the dispersibility of 0.1% FLU ophthalmic suspensions.** In the low (upper panels) and room (lower panels) temperature groups, each formulation of 0.1% FLU ophthalmic suspension was stored at 10 ˚C for three months and subsequently maintained at room temperature for two months or left at room temperature for five months. Optical microscopic images of the dispersible particles of FLU ophthalmic suspension at 100x magnifications are shown. P1; Original, P2; Generic A, P3; Generic B. All the images are representative of three independent experiments.

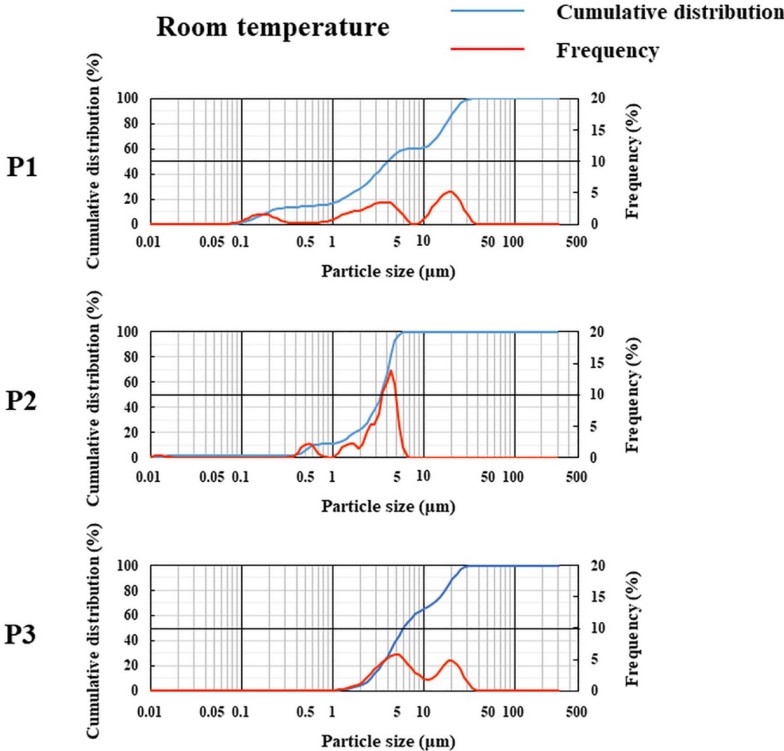

**Fig 5. Particle size distribution and cumulative frequency for each formulation of 0.1% FLU ophthalmic suspensions.** In the room temperature group, each formulation of 0.1% FLU ophthalmic suspension was stored at 10 ˚C for three months and subsequently maintained at room temperature for two months or left at room temperature for five months. Each panel represents the relative particle size distribution and cumulative frequency in three formulations of 0.1% FLU ophthalmic suspensions as measured on the laser diffraction particle size analyzer. P1; Original, P2; Generic A, P3; Generic B. All the images are representative of three independent experiments.

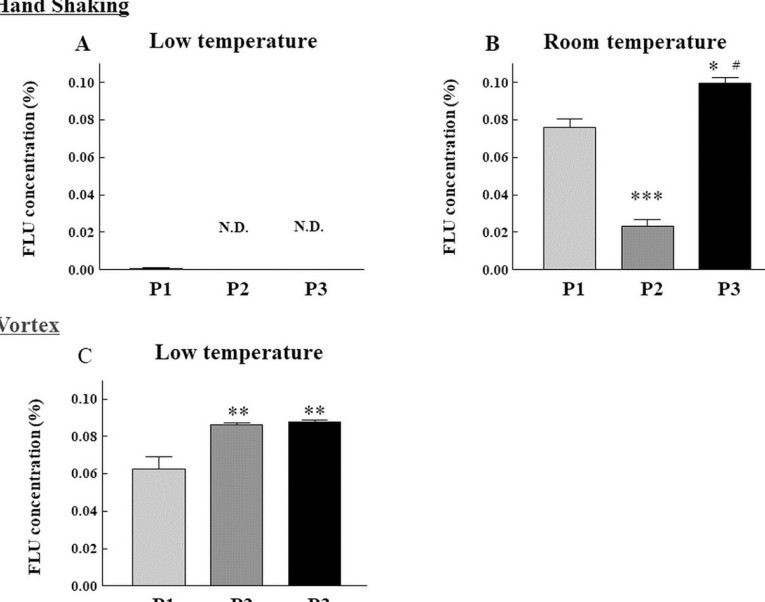

**Fig 6. Changes in the concentration of FLU ophthalmic suspensions contained in one drop following agitation by different methods.** In the low (left panel) and room (right panel) temperature groups, each formulation of 0.1% FLU ophthalmic suspension was stored at 10˚C for three months and subsequently maintained at room temperature for two months or left at room temperature for five months. The concentration of FLU suspensions contained in one drop from each formulation was measured by HPLC after agitation by hands (A, B) or using a vortex mixer (C). P1; Original, P2; Generic A, P3; Generic B, N.D.; not detected. n = 3, *** $p < 0.001$, ** $p < 0.01$, * $p < 0.05$ vs. Original, ### $p < 0.001$ vs. Generic A. All data are expressed as mean ± standard deviation and were analyzed by one-way analysis of variance followed by Tukey's test.

that in P2 and P3, and there were no differences between P2 and P3. In P1, the FLU content was approximately 63% of the standard value, 0.1%, whereas, in P2 and P3, it was as high as 86% to 88% of the standard value.

## Relationship between FLU particle dispersibility and physical properties in each sample

Fig 7 shows the viscosity in the low-temperature storage group. The viscosities at 10 ˚C and 25 ˚C from the graph are shown in Table 2. The viscosity of P2 was higher than those of P1 and P3. The following values for particles sedimented in P1, P2, and P3 in the low- and ordinary-temperature storage groups are shown in Table 2: $V_s/ (\rho_g - \rho_f)$ g. In both the low- and ordinary-temperature storage groups, $V_s/ (\rho_g - \rho_f)$ g, proportional to the terminal velocity, decreased in the following sequence: P3 $\gg$ P1 > P2. A comparison of the two groups showed higher values in the ordinary-temperature storage group. The zeta potential measurements for each eye-drop product are shown in Table 2. These results showed that the repulsive force between the particles in the different eye-drop products decreased in the following sequence: P2 > P3 $\gg$ P1.

## Discussion

Package inserts for eye-drops, except for a few products that are stored in an electrical refrigerator, recommend storing the product at room temperature, and the Japanese Pharmacopoeia Eighteenth Edition [14] defines room temperature as 1 to 30˚C and a cold place as 1 to 15˚C.

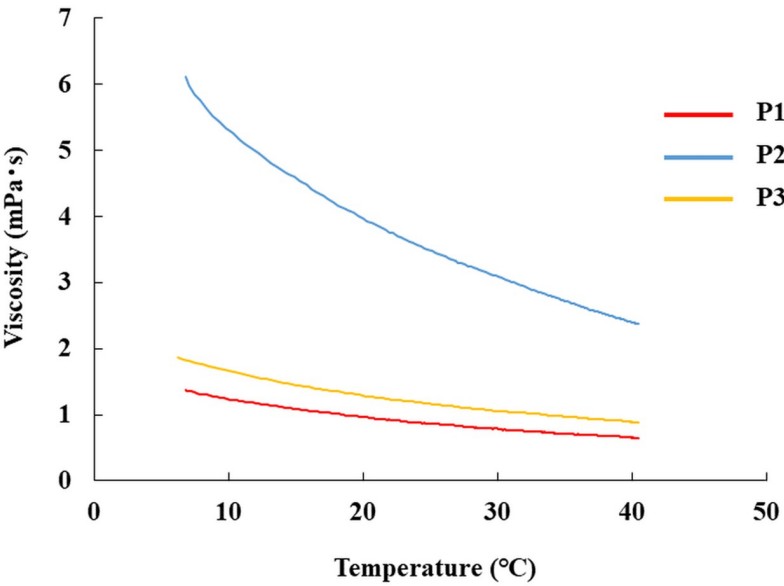

**Fig 7. Changes in the viscosity of 0.1% FLU ophthalmic suspensions of each formulation with temperature.** The viscosity of the 0.1% FLU ophthalmic suspension in each formulation was measured using a tuning-fork vibration-type viscometer at different temperatures. P1; Original, P2; Generic A, P3; Generic B.

For this reason, some clinicians and pharmacists recommend patients store eye-drops in an electrical refrigerator for preventing bacterial contamination and growth that may be induced by the storage of eye-drops at room temperature in Japan. Moreover, the temperature inside an electrical refrigerator for storing eye-drops is defined by the Japanese Industrial Standards as at least 0˚C and no higher than 8˚C, and the catalogs of most electrical appliance manufacturers state that the temperature range inside their refrigerators is ranged from 2 to 8˚C. Furthermore, during the Japanese winter (November to March), the mean temperatures in the living and changing rooms were 16.8˚C and 13.0˚C, respectively, while the mean minimum temperatures in the living room and changing room were 12.6˚C and 10.4˚C, respectively [15, 16]. This can also be lower in colder regions. Considering these factors, the storage temperatures for eye-drops are considered to range from low to ordinary temperatures. The storage temperature for FLU suspension eye-drops is also specified as 1–30 ˚C in Japan and the United States.

On the other hand, in Japan in 2012, Nitto Medic Co. released the following caution regarding the storage of a FLU eye-drop product: "Storage in cold places such as refrigerators should be avoided" [11]. Their product was a generic drug, and, as a means of maintaining the dispersibility of the principal drug for a specific time, polyvinyl alcohol is included as the usual thickening agent with methylcellulose (Table 1). Polyvinyl alcohol is known to cause irreversible gelling inside formulations at low temperatures [6–8], and it is considered that low-temperature storage will have the opposite effect to that intended to prevent the dispersion of the

**Table 2. Viscosity and zeta potential for 0.1% FLU ophthalmic suspensions used in this study.**

| Formulation | Particle size (μm) (mean±S.D.) | Viscosity (mPa · s) | | $V_s / (\rho_g - \rho_f) g$ (kg/m · s³) | | Zeta Potential (mV) |
|---|---|---|---|---|---|---|
| | | 10˚C | 25˚C | 10˚C | 25˚C | |
| P1 | 4.6±1.6 | 1.3 | 0.9 | $9.00 \times 10^{-10}$ | $13.06 \times 10^{-10}$ | -42.3 |
| P2 | 2.5±0.6 | 5.3 | 3.5 | $0.66 \times 10^{-10}$ | $0.99 \times 10^{-10}$ | -3.3 |
| P3 | 6.3±0.7 | 1.7 | 1.2 | $12.97 \times 10^{-10}$ | $18.38 \times 10^{-10}$ | -5.8 |

principal drug. Nevertheless, the authors could not find any warnings about the storage in refrigerators on the package inserts for any FLU eye-drop products in Japan and the United States. In the present study, to decide upon the cautions about the storage methods for suspension eye-drops to be made when giving instructions about their use, low-temperature storage was set as being at 10˚C, and the effects on dispersibility of the active component were investigated in the original drug and two generic drugs.

Firstly, when changes in external appearance with storage were investigated, sediment was found at the bases of all the eye-drop vials on completion of the storage period. Additionally, no difference in DTA was found between low- and ordinary-temperature storage with any of the eye-drop products, and no differences from FLU bulk powder were found. Therefore, it is considered that no changes occurred in the chemical compounds due to the storage conditions (Fig 3).

Next, we demonstrated the re-dispersibility of P1-P3 by agitation with hands. In the low-temperature storage group, the sediment was not eliminated after shaking the container by hand (Fig 2). Furthermore, the degree to which FLU particles in the eye-drop product were dispersed by shaking by hand was assessed using light microscopy, and almost no particles were found (Fig 4). Furthermore, an investigation using a laser diffraction particle size analyzer did not detect drug particles in the low-temperature storage group (Fig 5). Simultaneously, almost no FLU was detected in any of the samples of eye-drops (Fig 6). These findings showed that no dispersion occurred with ordinary shaking by hand

As against the above results, the present authors have previously reported that most of the sediment was eliminated by shaking by hand after short-term and long-term storage at the ambient temperature in the months of June to August [1]. This suggests that storage at low temperatures affects the dispersion of the sediment. In this manner, the authors have reported that it is important to explain, in the instructions for use, that if eye-drops are stored at an ambient temperature, shaking them vigorously is important before use to disperse them to a specified concentration. However, the present study performed a detailed investigation as to how safely eye-drops can be used after storage in a refrigerator. When each sample was shaken vigorously using a vortex after it failed to show dispersion by shaking by hand, the sediment at the base was dispersed to approximately 90% of the standard concentration in the two generic drugs, P2 and P3. In contrast, the concentration reached only approximately 60% of the standard value in P1, the original drug. Depending upon the product, no dispersion was achieved even with vigorous shaking using methods such as vortexing. We deduced two reasons to explain why P1 in the low-temperature storage group, the FLU content per drop did not closely approach the standard value even with both shaking by hand and shaking using a vortex. The first was dispersion into the bubbles formed by shaking with a vortex, and the second was the adhesion of the product to the vial walls; both effects were investigated. To investigate the dispersion into bubbles theory, vigorous shaking with a vortex was performed the day before measurement, and shaking was performed with care taken to avoid bubble formation on the measurement date. Upon measuring, the FLU contents with ordinary-temperature storage were found to be 0.080%, while it was 0.069% in low-temperature storage. To assess adhesion to the vial wall, all the eye-drop product was emptied from the vial, and the vial was then washed with purified water, filled with methanol, and shaken for 1 hour. Subsequently, the FLU concentration in methanol, considered to be indicative of the level of adhesion to the vial wall, was measured and found to be 0.011%. Upon combining the FLU concentrations in the eye-drop product and methanol, the value was approximately the same for both bubble-less ordinary-temperature storage and the bubble-less low-temperature storage group. These results suggest that, in the case of the original drug, P1, it is considered that some of the active components adhere to the vial wall when stored in a cold place.

There have been numerous previous reports regarding the adhesion of drugs to plastic containers [17–20]; however, most were related to water-based drugs, and the present authors have not been able to find any reports about FLU, which has low solubility. One possible factor in the adhesion of P1 to the vial wall in the present study relates to the chemical composition of the vial. However, according to the questionnaire used for each of the drugs, the vials for P1 and P2 were composed of polyethylene, and that for P3 was composed of polypropylene. Therefore, the cause was not considered to be a difference in the vial composition. In the case of P1 alone, the cause of adhesion of the suspension components to the vial wall could have partly been due to long-term storage at low temperatures, and therefore, this remains an area for future research. Another possibility with tubular containers is that while shaking by hand is gentle and involves moving up and down, shaking by the vortex is vigorous and rotational. Further studies exploring the dispersion using these and other shaking methods are needed to improve the current understanding.

In this study, the causes of the low dispersibility of FLU eye-drops when stored at low temperatures, and the differences between suspension eye-drop products were ascertained using Stokes' equation. We found that with decreasing terminal velocity, the dispersibility and FLU concentration in the liquid decreased. It is considered that this may be because the viscosity of suspension eye-drops decreases at low temperatures, and FLU particles gradually sediment out, forming a dense layer of sediment at the base of the vial [21, 22] In addition, FLU particle diameter may affect the differences in dispersibility between eye-drop products. The original drug FLU eye-drop product is a formulation with increased dispersibility due to a formulation design that uses aggregates with methylcellulose as a dispersant, which controls the particle sedimentation speed and inclusion of an appropriate dispersant [22]. P2 is compounded with polyvinyl alcohol, and P3 with methylcellulose, like the original drug. Therefore, the possibility of the different dispersants affecting the suspension properties cannot be ruled out. In addition, it is considered that P2, with low dispersibility, has the lowest zeta potential, and the low repulsive force between particles may also affect dispersibility.

## Conclusion

The results of our study demonstrated that when these products were stored at low temperature until use, such as in the refrigerator, ordinary shaking did not help achieve dispersion to the specified concentration, and even after vigorous shaking with some formulations, the specified concentration was not achieved due to bubble formation and/or adhesion to the vial's walls. Therefore, the instructions for the use of FLU eye-drops should be specified, with emphasis on the importance of the storage temperature. The importance of not storing at low temperatures, especially not in the refrigerator or a cold room, and shaking vigorously before use, should be stated in the instructions for patients. This study highlights the significance of recognizing the characteristic feature of FLU eye-drops and the information, especially for the storage temperature provided in the package insert. The findings are expected to help the clinicians and pharmacists involved in the ophthalmologic field to instruct patients on using and storing the FLU eye drops, i.e., mixing the drugs thoroughly before each use and storing them away from cold places including the refrigerators.

## Author Contributions

**Conceptualization:** Tokio Obata, Yuichiro Nakada.

**Data curation:** Saori Deguchi, Jyoji Yoshitomi, Kazunori Inaba, Noriaki Nagai.

**Formal analysis:** Tokio Obata, Saori Deguchi, Jyoji Yoshitomi, Kazunori Inaba, Yoko Urashima, Noriaki Nagai.

**Investigation:** Noriaki Nagai.

**Methodology:** Saori Deguchi, Jyoji Yoshitomi, Kazunori Inaba, Noriaki Nagai, Yuichiro Nakada.

**Project administration:** Kouichi Hosomi, Noriaki Nagai, Yuichiro Nakada.

**Supervision:** Yuichiro Nakada.

**Validation:** Saori Deguchi, Jyoji Yoshitomi, Kazunori Inaba, Noriaki Nagai.

**Visualization:** Tokio Obata, Yoko Urashima, Takuro Kobori.

**Writing – original draft:** Tokio Obata.

**Writing – review & editing:** Tokio Obata, Yoko Urashima, Takuro Kobori, Kouichi Hosomi, Noriaki Nagai, Yuichiro Nakada.

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
