## [Decision Letter · Decision Letter 0]

26 Sep 2022

PONE-D-22-18883Effect of Storage Temperature on the Dispersibility of commercially available 0.1% Fluorometholone Ophthalmic SuspensionPLOS ONE

Dear Dr. Nakada,

Thank you for submitting your manuscript to PLOS ONE. After careful consideration, we feel that it has merit but does not fully meet PLOS ONE’s publication criteria as it currently stands. Therefore, we invite you to submit a revised version of the manuscript that addresses the points raised during the review process.

ACADEMIC EDITOR: Two reviewers suggested for major revision and one reviewer suggested minor revision. ==============================

We look forward to receiving your revised manuscript.

Kind regards,

Ajaya Bhattarai

Academic Editor

PLOS ONE

Journal Requirements:

Reviewers' comments:

Reviewer's Responses to Questions

**Comments to the Author**

1. Is the manuscript technically sound, and do the data support the conclusions?

Reviewer #1: Yes

Reviewer #2: Yes

Reviewer #3: Yes

2. Has the statistical analysis been performed appropriately and rigorously? 

Reviewer #1: Yes

Reviewer #2: Yes

Reviewer #3: Yes

3. Have the authors made all data underlying the findings in their manuscript fully available?

Reviewer #1: Yes

Reviewer #2: Yes

Reviewer #3: Yes

4. Is the manuscript presented in an intelligible fashion and written in standard English?

Reviewer #1: Yes

Reviewer #2: No

Reviewer #3: Yes

5. Review Comments to the Author

Reviewer #1: This manuscript describes the dispersion behaviors of three fluorometholone ophthalmic suspension eye drops (one original drug and two generic drugs) of 0.1% at different temperatures. My opinions and comments about this manuscript are as follows:

1) In all manuscript, section numbers should be checked. The use of section numbers in only subsections is inappropriate.

2) Lines 91-93, etc.: In all manuscript and in references, the use of capital letters should be checked.

3) Line 111: “Further,” should be replaced by “Furthermore,”.

4) Lines 120, 131, 146-147, etc.: In all manuscript, there is no coherence in the use of capital and small letters during the spelling of subsection titles.

5) Lines 162, 164, 165, 170, etc.: In all manuscript, please use subscript for “s” in “Vs”, for “p” in “Dp”, for “g” in “ρg” and for “f” in “ρf”.

6) Lines 218, 251, etc.: In all manuscript, please use small letters in the spelling of “Low” and “Room”.

7) Lines 222-223; 255-256; 275-278; 300-301; etc.: The place of these statements is inappropriate!!! They should be given in figure captions, not in text.

8) Line 418: Please add a “Conclusion” section.

9) Line 419: “Reference” should be replaced by “References”.

10) Lines 419-492: This section should be checked according to the reference format of plos One.

11) Figure captions should be presented before figures.

12) Fig. 3: The quality of Fig. 3 should be improved.

13) Fig. 5:

• “frequency” should be replaced by “Frequency”.

• In x-axes, “Particle Size” should be replaced by “Particle size”.

Reviewer #2: The manuscript, although interesting and well-designed and conducted, needs extensive language editing. Some sentences are exceedingly long and difficult to follow. I suggest the authors to omit numbering in the results section and use plain subheading; or alternatively without any subheadings. If subheadings are preferred a similar style may be useful in the Discussion section as well.

Reviewer #3: An interesting and relevant topic.

It is however a very technical manuscript which is not so easy to read for the clinician. This is partly caused by the often long sentences with significant sub-sentences.

I have no questions concerning the content. The research is soundly executed, and is presented and discussed step by step. I do recommend shortening of the often complex and long sentences to increase the readability of this technical manuscript. Furthermore a suggestion to perhaps incorporate advices for the ophthalmic clinician, i.e. apart from the frequency of administration, a storage or use advice may also be discussed.

6. PLOS authors have the option to publish the peer review history of their article (what does this mean?). If published, this will include your full peer review and any attached files.

Reviewer #1: No

Reviewer #2: **Yes: **Eray Atalay

Reviewer #3: **Yes: **H.M. van Minderhout, MSc

---

## [Author Response · Author response to Decision Letter 0]

17 Oct 2022

Response to Reviewer 1’s Comments

We would like to thank #Reviewer 1 for the valuable suggestions on our manuscript. We have carefully read all of your comments and suggestions and have made the corrections in the revised version of the manuscript. Detailed responses to your comments are listed below, and we have highlighted all changes which can be viewed as tracked changes in the file labeled “Revised Manuscript with Track Changes.” We hope that the revised manuscript would be satisfactory for publication in PLoS ONE.

Q1. In all manuscript, section numbers should be checked. The use of section numbers in only subsections is inappropriate.

A1. According to #Reviewer 1’s comment, we removed the numbering in the Results section (Lines 188–189, 200–201, 223–234).

Q2. Lines 91-93, etc.: In all manuscript and in references, the use of capital letters should be checked.

A2. The reviewer’s comments are very important. We checked the use of capital letters in this manuscript. The name of products and vendors are not changed, since these are original notation. Thank you for pointing out this (Lines 91–93).

 

Q3. Line 111: “Further,” should be replaced by “Furthermore,”.

A3. Thank you very much for pointing this out. We revised to “Furthermore” from “Further” in Materials and Methods (Lines 111). 

Q4. Lines 120, 131, 146-147, etc.: In all manuscript, there is no coherence in the use of capital and small letters during the spelling of subsection titles.

A4. The reviewer’s comment is correct. We unified the use of capital and small letters in the spelling of subsection titles in Materials and Methods and Results. Please refer to the comments in the file labeled “Revised Manuscript with Track Changes” (Lines 103,125,140,149,172,181,188-189).

Q5. Lines 162, 164, 165, 170, etc.: In all manuscript, please use subscript for “s” in “Vs”, for “p” in “Dp”, for “g” in “ρg” and for “f” in “ρf”.

A5. Thank you for pointing out this. We have incorporated subscript “s”, “p”, “g”, and “f” into Stokes’ equation (Lines 32–33, 155, 157-158, 163, 228–229, Table 2).

Q6. Lines 218, 251, etc.: In all manuscript, please use small letters in the spelling of “Low” and “Room”.

A6. In order to respond to the reviewer’s comment, we replaced “Low” and “Room” with “low” and “room” throughout the manuscript. Please refer to the comments in the file labeled “Revised Manuscript with Track Changes” (Lines 446, 455, 464, 474).

Q7. Lines 222-223; 255-256; 275-278; 300-301; etc.: The place of these statements is inappropriate!!! They should be given in figure captions, not in text.

A7. The reviewer’s comments are very important. In order to respond to the reviewer’s comment, we moved all Figure Captions after the References (Lines 429–488).

Q8. Line 418: Please add a “Conclusion” section.

A8. Thank you for pointing out this. We added the Conclusion section (Lines 345–359).

Q9. Line 419: “Reference” should be replaced by “References”.

A9. According to #Reviewer 1’s comment, we corrected to “ References” (Line 361).

Q10. Lines 419-492: This section should be checked according to the reference format of plos One.

A10. The reviewer’s comment is correct. We revised the reference format according to the submission guidelines of PLoS ONE (References, Lines 361–426).

Q11. Figure captions should be presented before figures.

A11. Thank you very much for pointing this out. In order to respond to the reviewer’s comment, we moved all Figure Captions after the References (Lines 429–488).

Q12. Fig. 3: The quality of Fig. 3 should be improved.

A12. In order to respond to the reviewer’s comment, we improved the quality of Figure 3 (Figure 3).

Q13. Fig. 5:• “frequency” should be replaced by “Frequency”. • In x-axes, “Particle Size” should be replaced by “Particle size”.

A13. The reviewer’s comment is correct. We revised “frequency” to “Frequency”, and “Particle Size” to “Particle size” in Figure 5 (Figure 5).

Thank you for great comments.

 

Response to Reviewer 2’s Comments

We would like to thank #Reviewer 2 for the valuable suggestions on our manuscript. We have carefully read all of your comments and suggestions and have made the corrections in the revised version of the manuscript. Detailed responses to your comments are listed below, and we have highlighted all changes which can be viewed as tracked changes in the file labeled ‘Revised Manuscript with Track Changes’. We hope that the revised manuscript would be satisfactory for publication in PLoS ONE.

Q1. The manuscript, although interesting and well-designed and conducted, needs extensive language editing. Some sentences are exceedingly long and difficult to follow.

A1. In order to respond to the reviewer’s comment, this manuscript has been re-edited by English editing service Editage (Job code: BCIO_1_3).

Q2. I suggest the authors to omit numbering in the results section and use plain subheading; or alternatively without any subheadings. If subheadings are preferred a similar style may be useful in the Discussion section as well.

A2. Thank you very much for pointing this out. We removed the numbering in the Results section, and applyied the plain subheadings (Lines 187–234).

Thank you for great comments. 

Response to Reviewer 3’s Comments

We would like to thank #Reviewer 3 for the valuable suggestions on our manuscript. We have carefully read all of your comments and suggestions and have made the corrections in the revised version of the manuscript. Detailed responses to your comments are listed below, and we have highlighted all changes which can be viewed as tracked changes in the file labeled ‘Revised Manuscript with Track Changes’. We hope that the revised manuscript would be satisfactory for publication in PLoS ONE.

Q1. It is however a very technical manuscript which is not so easy to read for the clinician. This is partly caused by the often long sentences with significant sub-sentences. I have no questions concerning the content. The research is soundly executed, and is presented and discussed step by step. I do recommend shortening of the often complex and long sentences to increase the readability of this technical manuscript. Furthermore a suggestion to perhaps incorporate advices for the ophthalmic clinician, i.e. apart from the frequency of administration, a storage or use advice may also be discussed.

A1. According to #Reviewer 3’s comment, this manuscript has been re-edited by English editing service Editage (Job code: BCIO_1_3). We also incorporated the Conclusion section including the recommendations for ophthalmic clinicians and pharmacists (Lines 345–359).

Thank you for great comments.

---

## [Editor Report · Decision Letter 1]

25 Oct 2022

Effect of Storage Temperature on the Dispersibility of Commercially Available 0.1% Fluorometholone Ophthalmic Suspension

PONE-D-22-18883R1

Dear Dr. Nakada Yuichiro,

We’re pleased to inform you that your manuscript has been judged scientifically suitable for publication and will be formally accepted for publication once it meets all outstanding technical requirements.

Kind regards,

Ajaya Bhattarai

Academic Editor

PLOS ONE

Additional Editor Comments (optional):

The academic editor is satisfied with the revised manuscript.
---

## [Editor Report · Acceptance letter]

27 Oct 2022

PONE-D-22-18883R1 

Effect of Storage Temperature on the Dispersibility of Commercially Available 0.1% Fluorometholone Ophthalmic Suspension 

Dear Dr. Nakada:

I'm pleased to inform you that your manuscript has been deemed suitable for publication in PLOS ONE. Congratulations! Your manuscript is now with our production department. 

Kind regards, 

on behalf of

Dr. Ajaya Bhattarai 

Academic Editor

PLOS ONE